# Pretreatments of Non-Woody Cellulosic Feedstocks for Bacterial Cellulose Synthesis

**DOI:** 10.3390/polym11101645

**Published:** 2019-10-10

**Authors:** Ekaterina I. Kashcheyeva, Yulia A. Gismatulina, Vera V. Budaeva

**Affiliations:** Bioconversion Laboratory, Institute for Problems of Chemical and Energetic Technologies, Siberian Branch of the Russian Academy of Sciences (IPCET SB RAS), Biysk 659322, Altai Krai, Russia; julja.gismatulina@rambler.ru (Y.A.G.); budaeva@ipcet.ru (V.V.B.)

**Keywords:** non-woody cellulosic feedstocks, *Miscanthus*, oat hulls, pretreatment, enzymatic hydrolysis, bacterial cellulose

## Abstract

Pretreatment of biomass is a key step in the production of valuable products, including high-tech bacterial cellulose. The efficiency of five different pretreatment methods of *Miscanthus* and oat hulls for enzymatic hydrolysis and subsequent synthesis of bacterial cellulose (BC) was evaluated herein: Hydrothermobaric treatment, single-stage treatments with dilute HNO_3_ or dilute NaOH solution, and two-stage combined treatment with dilute HNO_3_ and NaOH solutions in direct and reverse order. The performance of enzymatic hydrolysis of pretreatment products was found to increase by a factor of 4−7. All the resultant hydrolyzates were composed chiefly of glucose, as the xylose percentage in total reducing sugars (RS) was 1−9%. The test synthesis of BC demonstrated good quality of nutrient media prepared from all the enzymatic hydrolyzates, except the hydrothermobaric treatment hydrolyzate. For biosynthesis of BC, single-stage pretreatments with either dilute HNO_3_ or dilute NaOH are advised due their simplicity and the high performance of enzymatic hydrolysis of pretreatment products (RS yield 79.7−83.4%).

## 1. Introduction

The comprehensive conversion of lignocellulosic biomass into a wide range of competitive products and energy through chemical and/or biotechnological techniques is a current fundamental trend in industrial biotechnology. Annually renewable plant raw materials are utilized to produce individual compounds such as monosaccharides, alcohols, acids, and monomers of biodegradable polymers for the chemical industry, power industry, and medicine [1,2,3,4,5,6,7,8]. One of the biorefining mainstreams is the development of technological fundamentals for biocatalysis of plant resources into glucose hydrolyzates—a universal nutrient broth for microbiological synthesis of various in-demand products, including BC (bacterial cellulose) [9,10,11,12]. The yield of reducing sugars from hydrolysis of native feedstocks is poor; therefore, one of the key problems is to devise efficient pretreatment methods that affect the structure, chemical composition, and hydrolysis rate of pretreated biomass [4,13,14,15]. A variety of physical (comminution, hydrothermolysis), chemical (acid, alkali, ozone), physicochemical (steam explosion, ammonia fiber explosion), and biological techniques have been developed for the pretreatment of lignocellulosic biomass [16,17,18,19,20].

The utilization of non-woody feedstocks as a source of cellulose, for example, energy crops or agricultural residues, is increasingly receiving interest among researchers [21,22,23]. *Miscanthus* and oat hulls are esteemed as promising, readily renewable sources of cellulose. *Miscanthus* is an energy crop unpretentious to breeding conditions, with a biomass gain of up to 15 ton/ha [24,25]. Oat hulls are an abundant and available raw source in agricultural regions worldwide, including Russia, and are basically regarded as pentose-rich biomass [26,27].

The conversion of different *Miscanthus* species such as *Miscanthus sinensis*, *Miscanthus giganteus,* and *Miscanthus sacchariflorus* into many valuable products, particularly bioethanol and paper, is being actively studied [24,28,29,30,31,32,33,34]. For *Miscanthus*, various pretreatment methods have been tested in order to obtain substrates. These methods employ sulfuric acid/ethanol/water, formic acid/acetic acid/water, formic acid/hydrogen peroxide/water, aqueous NaOH, ethylenediamine/DMSO, ethylenediamine/1-butyl-3-methylimidazolium dimethylphosphate, 1-ethyl-3-methylimidazolium acetate with ammonia and/or oxygen, autohydrolysis in water with and without 2-naphthol, ozone/ethanol and electrolyzed water with or without alkaline peroxide, as well as aqueous ammonia with or without hydrogen peroxide [35], various alkali and acids [34,36,37], concentrated sodium benzoate solution [38,39], and steam explosion [40].

From among the pretreatments listed above, the most attractive are those that use simple and available reagents or just water. In this paper, the following pretreatment approaches for *Miscanthus* and oat hulls are discussed: Hydrothermobaric treatment, single-stage treatment with dilute HNO_3_ or dilute NaOH, and two-stage treatment with HNO_3_ and NaOH in the direct and reverse sequence.

Hydrothermobaric treatment is viewed as one of the most efficient and low-cost pretreatment techniques, which improves the reactivity of substrates through means of steam diffusion into the plant cell wall to hydrolyze hemicelluloses and transform lignin at high temperature [41,42].

The application of dilute HNO_3_ and/or NaOH solutions, either combined (in different sequences) or separately, to process plant biomass offers a range of advantages. Two-stage pretreatment methods are highly effective in degrading non-cellulosic components of plant biomass to furnish substrates (cellulose) with a low content of non-cellulosics, which implies efficient enzymatic hydrolysis. The advantage of nitric acid for biomass treatment is that HNO_3_ exhibits an exceptional reactivity to lignin, allowing fast oxidative delignification at moderate temperature and atmospheric pressure [22,43,44]. Alkaline treatment of feedstocks can dissolve non-woody lignin at a small alkali concentration and atmospheric pressure [3].

It is evident that the application of simple reagents (or without), efficient pretreatment methods, and available feedstocks can afford high-reactive substrates, which will considerably reduce the cost of high-tech products, specifically BC.

BC is a type of nanopolymer that represents a nanostructured material with unique properties (nanoscale, renewable, low toxic, biocompatible, biodegradable, etc.) and wide applicability [12,45,46,47,48]. Over the last 20 years, there have been lab-scale experiments performed on synthesizing BC in different unconventional (inexpensive) nutrient media. The application of agricultural and industrial wastes has been studied as an approach to enhance the BC yield and reduce the costs of BC production [49]. These wastes include fruit processing residues, plant extracts, molasses, syrups, candy production wastewater, and plant biomass-derived hydrolyzates. There are reports investigating hydrolyzates derived from corn cob, elephant grass, spruce, wheat straws, fiber waste sludge, hot water extracted wood, cotton-based textile wastes, sweet potato pulp, and corn starch [47,49,50]. Thus, we propose herein *Miscanthus* and oat hulls may serve as potential sources of carbon for the synthesis of BC.

The present study aimed at assessing different pretreatment methods of *Miscanthus* and oat hulls for enzymatic hydrolysis and further synthesis of bacterial cellulose.

## 2. Materials and Methods

### 2.1. Feedstocks

The energy crop *Miscanthus* and the agricultural residues oat hulls were used as feedstocks in this study. *Miscanthus* and oat are widely spread worldwide and refer to fast-growing cereals that grow up within a short vegetative period of only six months. *Miscanthus* straw (aboveground part) and oat hulls are promising nonfood sources of BC.

*Miscanthus sacchariflorus* Andersson was planted in 2011 and grown on a proprietary plot of land in Biysk. Oat hulls were acquired from the Biysk Elevator company (the 2017 harvest, Biysk, Russia).

*Miscanthus* was chopped to a size of at most 10 mm prior to pretreatment. Oat hulls required no additional grinding, as they have naturally homogeneous morphology. Before pretreatment, oat hulls were rinsed with water (50–60 °C) to remove hulling bran and then dried to 9−10% moisture.

### 2.2. Pretreatments

#### 2.2.1. Hydrothermobaric Treatment (HTBT)

The feedstocks were treated by hydrothermobaric process in a high-pressure reactor [42]. The pre-ground (*Miscanthus* only) and wetted feedstocks were put into a 0.44-L cylindrical reactor fitted with an electric heating element outside. The reactor contents were heated to a specified temperature for a specified retention time. After the retention time was over, the reactor contents were transferred to a receiver tank and then washed with distilled water, squeezed, and dried. Each feedstock was treated eight times under the same conditions for comparative purpose and for accumulation of the target product. The load material weight per one run was 15 g on an oven-dry basis.

The HTBT conditions were as follows: Pressure 1.5 MPa, temperature 196−197 °C, and time 600 s. The yields on an oven-dry basis were 66% for *Miscanthus* and 44% for oat hulls.

#### 2.2.2. Dilute Nitric-Acid Treatment (DNAT)

The dilute nitric-acid treatment of *Miscanthus* and oat hulls was performed in a 250 L vessel at a solid-to-liquid ratio of 1:15 (*w*/*v*) under pilot production conditions. The load material weight was 10 kg. The feedstock was treated with a 3−6 wt% HNO_3_ solution at 90−95 °C for 10−12 h. The yields on an oven-dry basis were 38.7% for *Miscanthus* and 35.3% for oat hulls.

#### 2.2.3. Alkaline Delignification (AD)

Alkaline delignification of *Miscanthus* and oat hulls was run in a 250 L vessel at a solid-to-liquid ratio of 1:15 (*w*/*v*) under pilot production conditions. The load material weight was 10 kg. The feedstock was treated with a 3−6 wt% NaOH solution at 90−95 °C for 6−8 h. The yields on an oven-dry basis were 44.8% for *Miscanthus* and 41.1% for oat hulls.

#### 2.2.4. Nitric-Acid Pulping Method (NAPM)

The nitric-acid pulping of *Miscanthus* and oat hulls was carried out in a 250 L vessel. The load material weight was 10 kg. In this pretreatment method, the feedstocks were processed successively with dilute HNO_3_ and NaOH solutions as follows: Pre-hydrolysis with a 0.2–0.4 wt% HNO_3_ solution at 90–95 °C for 1 h in a solid-to-liquid ratio of 1:15 (*w*/*v*); treatment with a 3–6 wt% HNO_3_ solution at 90–95 °C for 10–12 h in a solid-to-liquid ratio of 1:15 (*w*/*v*); treatment with a 3–6 wt% NaOH solution at 90–95 °C for 2–4 h in a solid-to-liquid ratio of 1:15 (*w*/*v*); treatment with a 0.5–1.0 wt% NaOH solution at 90–95 °C for 1–2 h in a solid-to-liquid ratio of 1:10 (*w*/*v*); and, finally, decationation (souring) by treatment with 0.5–1.0 wt.% HNO_3_ at 40–60 °C for 15–30 min in a solid-to-liquid ratio of 1:15 (*w*/*v*) to furnish the target cellulose. The yields on an oven-dry basis were 27.1% for *Miscanthus* and 23.2% for oat hulls.

#### 2.2.5. Combined Pulping Method (CPM)

The combined pulping of *Miscanthus* and oat hulls was conducted in a 250 L vessel. The load material weight was 10 kg. The difference of CPM from NAPM is that the feedstocks were treated with reagents in the reverse order; that is, the feedstocks were processed first with dilute NaOH and then with dilute HNO_3_ as follows: Pre-hydrolysis with a 0.2–0.4 wt% HNO_3_ solution at 90–95 °C for 1 h in a solid-to-liquid ratio of 1:15 (*w*/*v*); treatment with a 3–6 wt% NaOH solution at 90–95 °C for 6–8 h in a solid-to-liquid ratio of 1:15 (*w*/*v*); treatment with a 3–6 wt% HNO_3_ at 90–95 °C for 4–6 h in a solid-to-liquid ratio of 1:10 (*w*/*v*); and washing the resultant product successively with 1 wt% NaOH and 1 wt% HNO_3_. The yields on an oven-dry basis were 37.5% for *Miscanthus* and 34.2% for oat hulls.

All the substrates obtained by HTBT, DNAT, and AD pretreatments, as well as celluloses obtained by NAPM and CPM pretreatments, were washed until neutral wash waters and dried at room temperature to 7–10% moisture.

### 2.3. Analysis of Feedstocks and Pretreatment Products

The feedstocks and their pretreatment products were characterized by standard analytical techniques as below.

The content of Kürschner cellulose was measured by extraction with mixed HNO_3_/alcohol in a ratio of 1:4 for 4 h. The molarity of HNO_3_ solution was 3.14 mol/L [51]. The α-cellulose content in the celluloses obtained by the NAPM and CPM pretreatments was determined by a method whereby cellulose is treated with a 17.5 wt% NaOH solution (45 mL) for 45 min and the undissolved residue is quantified after washing with 9.5 wt% NaOH and water, and then dried [52]. Klason lignin (acid-insoluble) and acid-soluble lignin were measured pursuant to TAPPI T222 om-83 [53]. Pentosans were transformed in boiling 13 wt% HCl solution to furfural, which was collected in the distillate and determined on a xylose-calibrated UNICO UV-2804 spectrophotometer (630 nm wavelength, United Products & Instruments, Inc., Dayton, NJ, USA) with the orcinol-ferric chloride reagent. The ash content was quantified by incinerating cellulose at 600 °C for 3 h [54]. Cellulose degree of polymerization (DP) was determined by the outflow time of cellulose solution in cadoxene (cadmium oxide in ethylenediamine) from a VPZh-3 viscometer with a capillary diameter of 0.92 mm.

All experiments were done in triplicate and data were expressed as average values.

### 2.4. Enzymes and Hydrolysis

Enzymatic hydrolysis of pretreatment products was performed with enzymes CelloLux-A (Sibbiopharm Ltd., Berdsk, Russia) and BrewZyme BGX (Tarchomin Pharmaceutical Works Polfa S.A., Warszawa, Poland) standardized against cellulase, xylanase, and *β*-glucanase activities (Table 1). The enzyme cocktail was injected as follows: CelloLux-A 40 FPU/g solid and BrewZyme BGX 15 FPU/g solid. The hydrolysis was run in 0.5 L cone flasks. The samples containing 5 g dry matter were put into the flask and poured with 0.1 L acetate buffer (pH = 4.7, 0.5 M). The enzymes dissolved in 0.05 L acetate buffer were then added. The flasks were capped and placed onto a PE-6410M horizontal shaker. Samples of 0.002 L in volume were collected every 8 h to evaluate the concentration increment of reducing sugars in the hydrolyzate. The reaction mass was filtered after 72 h to give a ready-to-use hydrolyzate and a solid residue of the unreacted substrate. Three samples of one substrate type were hydrolyzed at a time to obtain accurate results.

### 2.5. Analysis of Hydrolyzates

The concentration of reducing sugars (RS) expressed as glucose in the hydrolyzate was measured spectrophotometrically (UNICO UV-2804 spectrophotometer, United Products & Instruments, Inc., Dayton, NJ, USA) using the 3,5-dinitrosalicylic acid (DNS) reagent [55]. HPLC analysis of monosaccharides in the hydrolyzates was performed on a Milichrom A-02 microcolumn liquid chromatograph (EcoNova, Novosibirsk, Russia) using a Ø 2 × 75 mm column filled with ProntoSIL-120-5-c18 sorbent (Bischoff, Germany). This method is based on the derivatization reaction of hydrolyzate monosugars with 2,4-dinitrophenylhydrazine. Chromatographic conditions were as follows: 0.1% trifluoroacetic acid as eluent A and 0.1% trifluoroacetic acid and 70% acetonitrile as eluent B; gradient 2200 µL 16%−27% B, 200 µL 27%−100% B, 600 µL 100% B; eluent flow rate 150 µL/min, temperature 35 °C; 360 nm wavelength for detection; sample volume 5 µL.

The final RS yields were estimated on a substrate weight basis (Equation (1)) and on a hydrolyzables content basis (with the deduction of non-cellulosic impurities such as ash and lignin) (Equation (2)), and the xylose yield was calculated on a pentosan content basis (Equation (3)), as given below.

(1)$$\eta_{RS}=\frac{C_{F}\cdot V}{m_{S}}\cdot 0.9\cdot 100$$
where *η_RS_* is the RS yield on a substrate weight basis (%); *C_F_* is the final RS concentration in hydrolyzate (g/L); *V* is the hydrolyzate volume (L); 0.90 is the coefficient attributed to the water molecule addition to anhydroglucose residues of the corresponding monomer units as a result of enzymatic hydrolysis; *m_S_* is the substrate weight for fermentation (g).
(2)$$\eta_{RSH}=\frac{C_{F}\;\cdot V}{m_{S}\cdot \left(100-L-A\right)}\cdot 0.9\cdot 100\cdot 100.$$
where *η_RSH_* is the RS yield on a hydrolyzables content basis (%); *C_F_* is the final RS concentration in hydrolyzate (g/L); *V* is the hydrolyzate volume (L); 0.90 is the coefficient attributed to the water molecule addition to anhydroglucose residues of the corresponding monomer units as a result of enzymatic hydrolysis; *m_S_* is the substrate weight for fermentation (g); *L* is the lignin content in the substrate (%); *A* is the ash content in the substrate (%).
(3)$$\eta_{X}=\frac{C_{X}\;\cdot V}{m_{S}\cdot \;P}\cdot 0.88\cdot 100\cdot 100$$
where *η_X_* is the xylose yield on a pentosan content basis (%); *C_X_* is the xylose concentration in hydrolyzate (g/L); *V* is the hydrolyzate volume (L); 0.88 is the coefficient attributed to the water molecule addition to anhydroxylose residues of the corresponding monomer units as a result of enzymatic hydrolysis; *m_S_* is the substrate weight for fermentation (g); *P* is the pentosan content in the substrate (%).

The work was conducted on equipment of the Biysk Regional Center for Shared Use of Scientific Equipment (IPCET SB RAS, Biysk, Russia).

### 2.6. Biosynthesis of BC in Nutrient Broths of Enzymatic Hydrolyzates of Pretreatment Products

Enzymatic hydrolyzates obtained in aqueous medium under conditions described in Section 2.4 were used as nutrient broths for biosynthesis of BC. The pH during hydrolysis was adjusted with orthophosphoric acid and ammonia. The hydrolyzates were filtered and the glucose concentration was measured by HPLC. The *Medusomyces gisevii* Sa-12 symbiont was employed as the producer. Black tea (15 g/L) was added to the hydrolyzates heated to 100 °C, the mixture was cooled to room temperature, and tea residues were filtered off. For the producer used herein, the tea is a standard component of a nutrient medium [56] because tea extractives stimulate biosynthesis of BC [57]. The synthesis of BC was performed in 0.25 L flasks under static conditions optimum for a synthetic glucose medium: Temperature 27 °C and time seven days [58]. The nutrient medium volume was 0.1 L and the inoculum dosage was 10 vol% [11].

BC was washed with distilled water, dilute NaOH, and HCl solutions, and distilled water again, until complete removal of cell debris and black tea colorants.

The BC biosynthetic experiments were run in triplicate for each nutrient medium.

The BC network structure was examined by scanning electron microscopy (SEM) using a JSM-840 microscope with a Link-860 Series II X-ray microanalyzer (JEOL, Tokyo, Japan). For this, BC hydrogels were pre-dehydrated with ethanol, freeze-dried, fixed onto a conductive adhesive tape, and sputter-coated with 10 nm of silver for 2 min at 20–30 Å before observation [59].

Figure 1 depicts a general flowchart of research.

## 3. Results and Discussion

### 3.1. Effect of Pretreatment on Chemical Composition of Feedstocks

The chemical compositions of the feedstocks and their pretreatment products are displayed in Figure 2.

#### 3.1.1. Feedstocks

The comparison between the chemical compositions of the feedstocks (Figure 1) showed that *Miscanthus* contained 52.1% cellulose, 18.6% lignin, 21.3% pentosans, and 4.8% ash, whereas oat hulls comprised 44.7% cellulose, 18.1% lignin, 30.8% pentosans, and 4.6% ash. It is thus evident that the feedstocks were mainly composed of hydrolyzable components: 73–76% cellulose plus pentosans and 23% total non-hydrolyzables. The pretreatment methods under study would presumably be able to increase the cellulosic fraction of the substrates and thereby diminish the proportions of pentosans and non-hydrolyzables.

#### 3.1.2. Hydrothermobaric Treatment (HTBT)

The comparison between the chemical compositions of the untreated and hydrothermobarically treated feedstocks showed that the HTBT pretreatment diminished hydrolyzable hemicelluloses by 3.7−5.0 times, increased the cellulose content by 1.4−1.5 times, and increased the lignin content by 1.2−1.3 times in the resultant substrates.

The comparison between characteristics of the substrates obtained from the two feedstocks under the same conditions demonstrated that pretreated *Miscanthus* (HTBT *Miscanthus*) had a higher cellulose content than pretreated oat hulls (HTBT oat hulls) (72% vs. 65%), which can be explained by the fact that raw *Miscanthus* is richer in cellulose (57% vs. 47%). HTBT *Miscanthus* had a lower ash content of 2.9% compared to 7.3% for HTBT oat hulls, while pentosans were at the same level of 6%, irrespective of the feedstock type.

#### 3.1.3. Dilute Nitric-Acid Treatment (DNAT)

The DNAT pretreatment of the feedstocks broke down the lignocellulosic matrix into major constituents. It also decreased the hemicellulose content by 4.2−4.5 times and partially solubilized, oxidized, and nitrated lignin, thereby decreasing the lignin content in the substrate by 1.5−2.1 times. The cellulose content after treatment increased by 1.5−1.6 times in the substrate.

The comparison of pretreated *Miscanthus* and oat hulls showed that DNAT *Miscanthus* contained more cellulose, 80% vs. 72%, less lignin, 8.8% vs. 12.3%, and less pentosans, 5.8% vs. 7.4%.

#### 3.1.4. Alkaline Delignification (AD)

Alkaline delignification removed basically lignin. The post-treatment lignin content declined by 4.5–5.5 times. Moreover, the contents of hemicelluloses and ash decreased by 3.3–3.8 times and 1.4–1.8 times, respectively. The removal of non-cellulosics by alkaline delignification increased the cellulose content by 1.7–1.9 times.

#### 3.1.5. Nitric-Acid Pulping Method (NAPM)

In this method, the pre-hydrolysis step cleaved the lignocellulosic matrix into major constituents to partially remove hemicelluloses. Further treatment with dilute HNO_3_ completely removed hemicelluloses (to 1.7–2.0%), partially solubilized, oxidized, and nitrated lignin to form nitrolignin. The subsequent alkaline treatment transferred nitrolignin into the liquor and eliminated it from the product.

Such a treatment was able to reduce the lignin (by 18.6–22.6 times) and hemicellulose contents (by 12.5–15.4 times) and enhance the cellulose content in the product by 1.8–2.1 times.

The NAPM method afforded a high-quality product with *α*-cellulose contents of 93.5% for *Miscanthus* and 94.0% for oat hulls, and total non-hydrolyzables were at most 3.3%. The cellulose degree of polymerization was almost similar: 1100 for NAPM *Miscanthus* and 1140 for NAPM oat hulls.

#### 3.1.6. Combined Pulping Method (CPM)

The combined pulping method at the first step degraded the lignocellulosic matrix and removed fat-soluble and water-soluble matters and partially hemicelluloses. Then, the dilute NaOH treatment removed lignin. The subsequent HNO_3_ treatment removed hemicelluloses and extracted undissolved lignin residues. Such a pretreatment could decrease the lignin (by 13.3–36.2 times) and hemicellulose (в 3.3–4.5 times) contents and raise the cellulose content in the product by 1.8–2.1 times.

The cellulose samples thus obtained had 91.5% (*Miscanthus*) and 92.5% (oat hulls) α-cellulose, 6.4–6.9% pentosans, and at most 2.1% lignin plus ash for *Miscanthus* and 0.6% for oat hulls. The cellulose degree of polymerization was 1030 for CPM *Miscanthus* and 1150 for CPM oat hulls.

It should be noted that the comparative analysis between the chemical compositions of all the pretreatment products did not allow us to identify the leader in enzymatic hydrolysis and predict the efficiency results.

We believe that the change in chemical composition of substrates after chemical pretreatment is determinant for the efficiency of enzymatic hydrolysis. The comparison of the pretreatment methods (Figure 1) demonstrates an increase in the cellulose content in all cases: 72.3%−93.5% for Miscanthus and 65.1%−94.0% for oat hulls. However, only the substrate prepared by hydrothermobaric treatment (HTBT substrate) exhibits an increment in the lignin content after the HTBT treatment; hence, the lignin role as the barrier remains the same as for the untreated feedstock.

### 3.2. Enzymatic Hydrolysis of Pretreated Biomass

Enzymatic hydrolysis is a way of assessing the pretreatment efficiency of feedstocks. We therefore examined the reducing sugar (RS) concentration as a function of enzymatic hydrolysis time. The time course of RS concentration during enzymatic hydrolysis of the substrates under study is illustrated in Figure 3. The hydrolyzates obtained in 72 h of hydrolysis were characterized and the results are summarized in Table 2.

#### 3.2.1. Enzymatic Hydrolysis of Feedstocks

As it follows from Figure 3 and Table 2, the native feedstocks exhibited the poorest reactivity, as expected. The enzymatic hydrolysis of the untreated feedstocks took place slowly, but oat hulls were converted a bit better than *Miscanthus*. The hydrolysis was observed to proceed at a high rate for the initial 16 h to accumulate 50–70% of reducing sugars (RS) whose concentration reached 2.6 g/L for *Miscanthus* and 3.2 g/L for oat hulls. The process then slowed down and the RS concentration in 72 h attained 4.1 g/L for *Miscanthus* and 4.5 g/L for oat hulls, corresponding to the yields of 11% and 12% on a solid weight basis, respectively. Carbohydrates were incompletely transformed to monosaccharides, which was probably due to poor enzyme access to the substrates.

#### 3.2.2. Enzymatic Hydrolysis of HTBT Substrates

The treatment of the feedstocks in a high-pressure reactor under 1.5 MPa at 196−197 °C for 600 s raised the substrate conversion degree by a factor of 4.0 for HTBT *Miscanthus* and by a factor of 5.6 for oat hulls. The RS yield after 72 h of hydrolysis was 44% for HTBT *Miscanthus*, which is 1.5 times lower than that for HTBT oat hull (68%). However, the hydrolysis rates for HTBT *Miscanthus* and HTBT oat hulls were almost identical for initial 8 h of hydrolysis: The RS concentrations were 6.2 g/L and 7.0 g/L, respectively. The hydrolysis of HTBT oat hulls further kept going at the same rate, with the RS concentration reaching 23 g/L in 32 h (RS yield 62%), whereupon it decelerated. By that time, the RS concentration curve for HTBT *Miscanthus* achieved a plateau at 15 g/L. Given that the both substrates were alike in chemical composition, the observed divergence could only be attributed to their different natural structures.

HTBT *Miscanthus* provided RS yields of 44% on a substrate weight basis and 59% on a hydrolyzables basis, with the xylose percentage in total RS being 9%. HTBT oat hulls obtained under the same conditions exhibited a higher reactivity to enzymatic hydrolysis and afforded RS yields of 68% on a substrate weight basis and 98% on a hydrolyzables basis, and the xylose proportion was only 6%, which is 1.5 times lower. The hydrolysis of HTBT oat hulls would guarantee the obtention of a glucose-rich hydrolyzate. The comparison of HPLC results for glucose and xylose in the HTBT oat-hull hydrolyzate (Table 2) corroborates the above assumption. Because both substrates had similar chemical compositions, the better reactivity of HTBT oat hulls towards enzymatic hydrolysis might be attributed to the physical nature of this feedstock representing fine plates of lignocellulose whose physicochemical treatment produces ribbon-like fibers that are easily cleaved lengthwise. After the hydrolysis was completed and the hydrolyzates were filtered, the residue from HTBT oat hulls represented only powdered lignin-like substances in contrast to hard HTBT *Miscanthus* particles that held the shape after hydrolysis.

#### 3.2.3. Enzymatic Hydrolysis of DNAT Substrates

The single-stage dilute nitric-acid treatment (DNAT) enhanced the reactivity to enzymatic hydrolysis by 7.3 times for *Miscanthus* and by 6.6 times for oat hulls and provided the same RS concentration of 29.5–29.8 g/L (RS yield 79.7–80.5% on a substrate weight basis) in hydrolyzates in 72 h. Alongside, the kinetic plot for DNAT *Miscanthus* showed a high initial hydrolysis rate: Half of the final RS concentration (15.2 g/L, yield 41.0% on a substrate weight basis) accumulated for initial 8 h of hydrolysis and a further increase in RS concentration almost ceased in 32 h. DNAT oat hulls showed a lower initial hydrolysis rate (RS concentration 10.0 g/L, yield 27% for initial 8 h) and a slow increase in RS concentration during hydrolysis. Since DNAT *Miscanthus* contained more hydrolyzables than DNAT oat hulls, the RS yields were 92.0% and 98.4% on a hydrolyzables basis, respectively, and the xylose proportions in total RS were 3.4% and 6.8%, respectively.

#### 3.2.4. Enzymatic Hydrolysis of AD Substrates

The alkaline delignification (AD) with dilute NaOH enhanced the reactivity to enzymatic hydrolysis, irrespective of the feedstock type, by 7.2 times for *Miscanthus* and by 6.9 times for oat hulls, providing the following final RS concentrations in 72 h: 29.6 g/L for AD *Miscanthus* and 30.9 g/L for AD oat hulls, which is equivalent to final RS yields of 79.9% and 83.4% on a substrate weight basis and of 85.0% and 90.0% on a hydrolyzables basis, respectively. The xylose percentages in total RS were 5.7% for AD *Miscanthus* and 6.5% for AD oat hulls.

It is obvious that a similar tendency is observed for the substrates obtained by the single-stage DNAT and AD pretreatments: The *Miscanthus* substrates were distinguished by a sharp increase in RS concentration nearly to the maximum for initial 24–32 h of hydrolysis, while the oat hull substrates exhibited a slower RS accumulation to attain a maximum RS concentration only in 56–64 h of hydrolysis.

By intercomparing the hydrolysis results for the single-stage pretreatment products, it becomes evident that the maximum increase in reactivity of the feedstocks is provided by HNO_3_ or NaOH treatment: In all cases, the final RS concentration in the hydrolyzates was 29.5–30.9 g/L, which is equivalent to the RS yield of 79.7–83.4% on a substrate weight basis. The RS yield of 85.0–98.4% on a hydrolyzables basis and the xylose yield of 46.6–73.0% on a pentosan basis indicate a nearly complete conversion of carbohydrates to monosugars and emphasize the efficiency of the single-stage pretreatments. The obtained hydrolyzates were composed chiefly of glucose, with the xylose concentration being as low as 1.0–2.0 g/L.

#### 3.2.5. Enzymatic Hydrolysis of Celluloses

Celluloses obtained by the nitric-acid pulping method (NAPM celluloses) were 6.1–6.5 times more reactive than the feedstocks: The RS concentration was 26.7–27.5 g/L in 72 h, and the RS yield was 72.1–74.3% on a substrate weight basis. In hydrolysis of celluloses obtained by the combined pulping method (CPM celluloses), the final RS concentration increased by a factor of 6.8–7.3 compared to the feedstocks, providing the RS concentration of 30.0–30.8 g/L in the hydrolyzates and the RS yield of 81.0–83.2% on a substrate weight basis. The NAPM celluloses were less reactive than the CPM ones. In hydrolysis of the NAPM celluloses, the xylose concentration was as low as 0.2–0.3 g/L, which is commensurate with the xylose percentage of 0.7–1.1% in total RS. For the CPM celluloses, these parameters were slightly higher and constituted 0.8–0.9 g/L and 2.7–2.9%, respectively.

The obtained results demonstrate high performance of all the pretreatment methods under study and guarantee chiefly glucose hydrolyzates with a glucose concentration of 14.21−28.98 g/L. However, it is hard to identify a substrate of choice for the synthesis of BC. Even the HTBT *Miscanthus* hydrolyzate (glucose concentration 14.21 g/L) can be used to prepare a nutrient medium. Therefore, enzymatic hydrolysis was performed for each substrate in aqueous medium and the final glucose concentration was analyzed. The glucose concentrations of the aqueous hydrolyzates were found to be close to those of the acetate-buffer hydrolyzates (Table 3).

To ensure the same initial glucose concentration during biosynthesis of BC, all the hydrolyzates, except for HTBT *Miscanthus,* were adjusted by dilution with distilled water to 14.0 g/L and used as nutrient media.

### 3.3. Biosynthesis of BC in Nutrient Broths Prepared from Enzymatic Hydrolyzates

The results of the BC synthesis in nutrient media prepared from *Miscanthus* and oat-hull enzymatic hydrolyzates are summarized in Table 4.

All the nutrient media, except for HTBT *Miscanthus* and HTBT oat hulls, were found to produce BC hydrogels. After being washed, the BC samples represented pearl-white thin films 3 mm wide. The fact that no BC was formed in nutrient broths prepared from HTBT pretreatment products can be explained by inhibitors present in microquantities. These inhibitors probably had a toxic effect on the symbiont but did not influence the enzyme activity at the preceding hydrolysis stage. The reason that these compounds are formed is probably due to harsh HTBT pretreatment conditions for feedstocks [3,50]. During the enzymatic hydrolysis of the HTBT pretreatment products, these inhibitors passed into the hydrolyzate liquid phase and their activity during biosynthesis of BC was impossible to prevent.

To confirm the bacterial nature of the BC samples obtained in this study, scanning electron microscopy (SEM) was used and the results are given in Figure 4.

It is seen in Figure 4 that all the BC samples have a random network structure consisting of ribbons less than 100 nm wide, which is on a par with the literature data [10,47]. Most researchers believe that it is the network structure of BC that determines the unique properties of BC [60].

It has thus been found that all the chemical pretreatment methods used herein afford good-quality substrates for nutrient broths. For the multistep process of BC synthesis from *Miscanthus* and oat hulls, it is evident that the single-stage pretreatments of feedstocks are more preferable on account of reduced feedstock treatment time and total saving of reagents and power inputs.

The use of, firstly, simple reagents such as HNO_3_ and NaOH, secondly, dilute solutions thereof, and, thirdly, standard chemical equipment in the single-stage pretreatment will positively reflect on the industrial synthetic process of BC. The literature describes a good deal of potential feedstocks (food and nonfood), various pretreatment techniques and hydrolysis types (chemical and enzymatic) for BC synthesis [10,47,61]. The advantage of the biosynthetic strategy for BC suggested herein is that it uses available and nonfood feedstocks, simple chemical pretreatment, and mild and efficient hydrolysis conditions.

## 4. Conclusions

*Miscanthus* and oat hulls are promising feedstocks to prepare nutrient broths for the synthesis of BC. Except for the hydrolyzate obtained from the hydrothermobaric pretreatment product, all the nutrient broths afforded BC whose network structure was confirmed by SEM. From the perspective of economic expediency, single-stage pretreatment methods are of choice. The single-stage pretreatment with dilute HNO_3_ or dilute NaOH was found to enhance the reactivity of both feedstocks by 6.6−7.3 times: The enzymatic hydrolysis of the resultant substrates furnished hydrolyzates with a reducing sugar (RS) yield of 79.7–83.4% on a substrate weight basis, with the xylose content in total RS being at most 6.8%. The RS yield of 85.0–98.4% expressed as hydrolyzables, as well as the xylose yield of 46.6–73.0% expressed as pentosans in the substrate, evidences a nearly complete conversion of carbohydrates to monosugars, thereby proving high performance of the single-stage pretreatments.

## Figures and Tables

**Figure 1 polymers-11-01645-f001:**
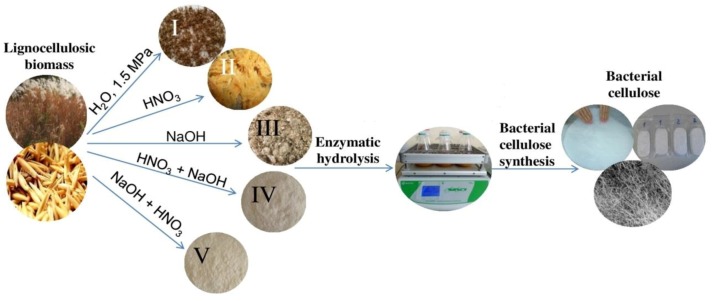
Research flowchart.

**Figure 2 polymers-11-01645-f002:**
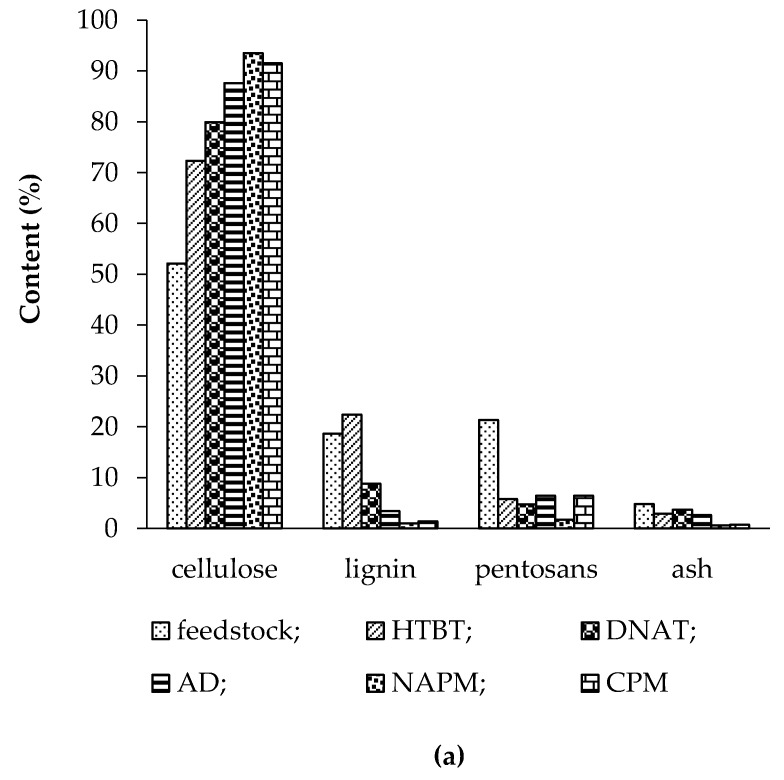
Cellulose, lignin, pentosan, and ash contents of native feedstocks and pretreatment products: (**a**) *Miscanthus* and (**b**) oat hulls.

**Figure 3 polymers-11-01645-f003:**
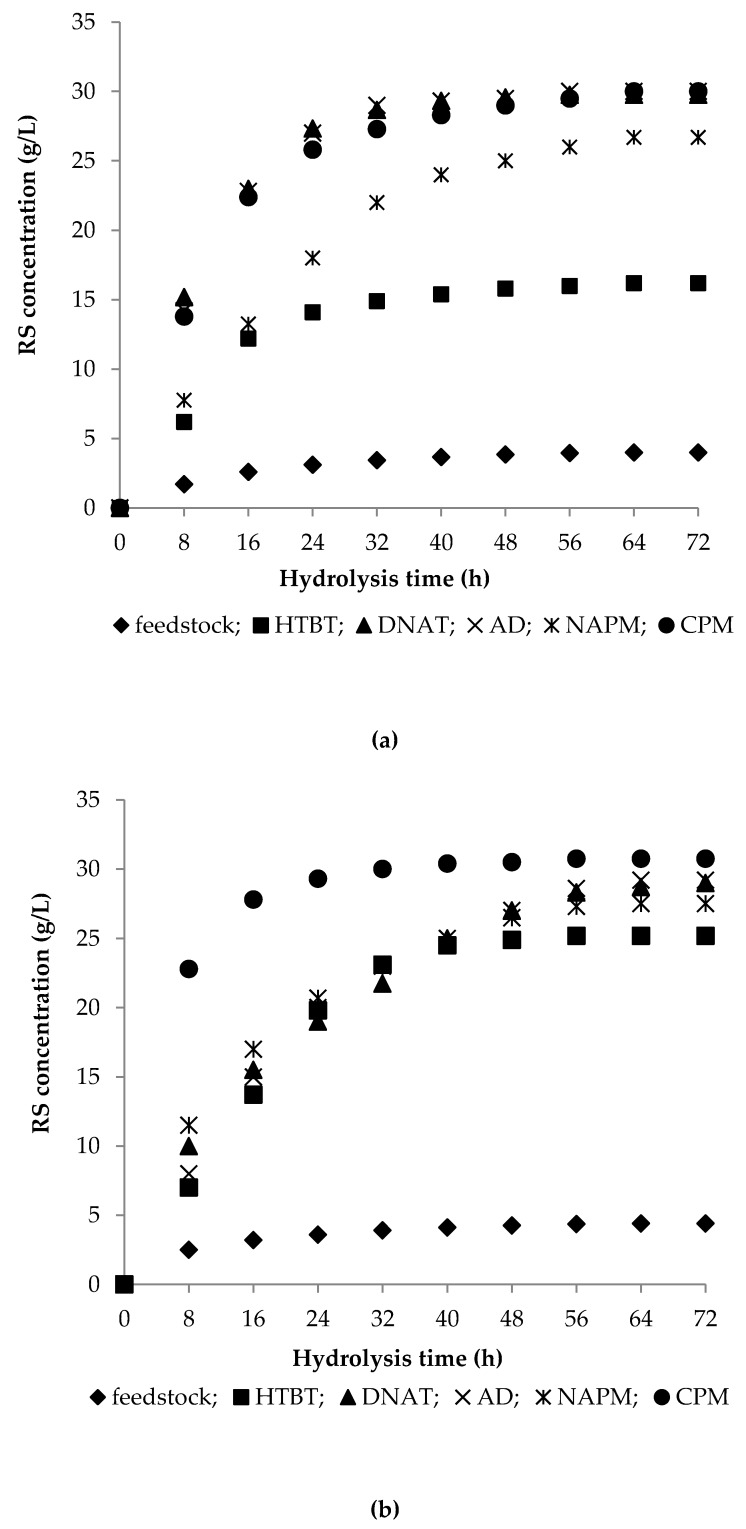
Time course of reducing sugars (RS) concentration in enzymatic hydrolysis of native feedstocks and pretreatment products: (**a**) *Miscanthus* and (**b**) oat hulls.

**Figure 4 polymers-11-01645-f004:**
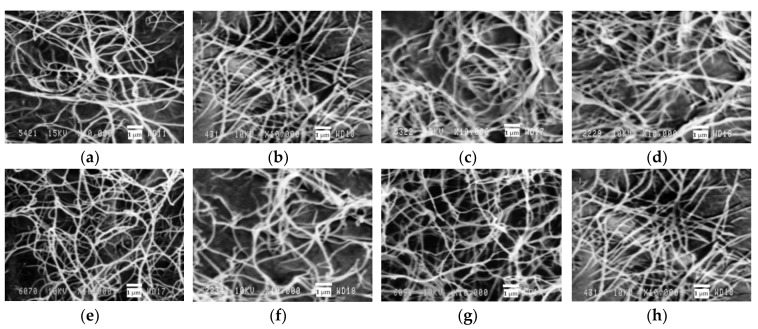
SEM images of BC samples produced in enzymatic hydrolyzates (×10,000 zoom): (**a**) dilute nitric-acid treatment (DNAT) *Miscanthus*, (**b**) DNAT oat hulls, (**c**) alkaline delignification (AD) *Miscanthus*, (**d**) AD oat hulls, (**e**) nitric-acid pulping method (NAPM) *Miscanthus* cellulose, (**f**) NAPM oat-hull cellulose, (**g**) combined pulping method (CPM) *Miscanthus* cellulose, (**h**) CPM oat-hull cellulose.

**Table 1 polymers-11-01645-t001:** Enzymes and their activities.

Enzyme	Enzymatic Activity
CelloLux-A (cellulose-standardized)	Cellulase: 2000 ± 10% CMCaseAU/g^a^ Xylanase: 8000 ± 10% XAU/g^b^ *β*-glucanase: 1500 ± 10% *β*-glAU/g^c^
BrewZyme BGX (hemicellulose-standardized)	Cellulase: 2100 ± 5% CMCaseAU/g Xylanase: 4200 ± 5% XAU/g *β*-glucanase: 530 ± 5% *β*-glAU/g

^a^ CMCaseAU/g—carboxymethylcellulase activity units per gram. ^b^ XAU/g—xylanase activity units per gram. ^c^
*β*-glAU/g—β-glucanase activity units per gram.

**Table 2 polymers-11-01645-t002:** Characteristics of hydrolyzates obtained after 72 h of enzymatic hydrolysis (acetate buffer).

Sample	Reducing Sugars			Concentration (g/L)	
	Conc. (g/L)	Yield (%)		Glucose	Xylose
		On Solid Weight Basis	On Hydrolyzables Basis		
*Miscanthus*	4.1 ± 0.1	11.1 ± 0.3	14.5 ± 0.4	3.9 ± 0.0	0.10 ± 0.0
Oat hulls	4.5 ± 0.1	12.2 ± 0.3	15.8 ± 0.4	4.3 ± 0.0	0.10 ± 0.0
HTBT *Miscanthus*	16.2 ± 0.1	43.7 ± 1.4	58.5 ± 1.5	14.2 ± 0.1	1.5 ± 0.1
HTBT oat hulls	25.2 ± 0.1	68.0 ± 1.4	98.1 ± 1.5	22.9 ± 0.1	1.5 ± 0.1
DNAT *Miscanthus*	29.8 ± 0.1	80.5 ± 1.4	92.0 ± 1.5	27.9 ± 0.2	1.0 ± 0.1
DNAT oat hulls	29.5 ± 0.1	79.7 ± 1.4	98.4 ± 1.5	26.6 ± 0.2	2.0 ± 0.1
AD *Miscanthus*	29.6 ± 0.1	79.9 ± 1.4	85.0 ± 1.5	27.0 ± 0.2	1.7 ± 0.1
AD oat hulls	30.9 ± 0.1	83.4 ± 1.4	90.0 ± 1.5	28.0 ± 0.2	2.0 ± 0.1
NAPM *Miscanthus* cellulose	26.7 ± 0.1	72.1 ± 1.4	73.3 ± 1.5	25.7 ± 0.2	0.2 ± 0.0
NAPM oat hull cellulose	27.5 ± 0.1	74.3 ± 1.4	75.5 ± 1.5	26.4 ± 0.2	0.3 ± 0.0
CPM *Miscanthus* cellulose	30.0 ± 0.1	81.0 ± 1.4	82.7 ± 1.5	28.3 ± 0.2	0.8 ± 0.1
CPM oat hull cellulose	30.8 ± 0.1	83.2 ± 1.4	83.7 ± 1.5	29.0 ± 0.2	0.9 ± 0.1

HTBT—hydrothermobaric treatment; DNAT—dilute nitric-acid treatment; AD—alkaline delignification; NAPM—nitric-acid pulping method; CPM—combined pulping method

**Table 3 polymers-11-01645-t003:** Glucose concentrations in hydrolyzates obtained after 72 h of enzymatic hydrolysis (aqueous medium).

Sample	Glucose Concentration (g/L)
HTBT *Miscanthus*	14.0 ± 0.1
HTBT oat hulls	22.2 ± 0.1
DNAT *Miscanthus*	26.9 ± 0.2
DNAT oat hulls	25.8 ± 0.2
AD *Miscanthus*	26.3 ± 0.2
AD oat hulls	27.1 ± 0.2
NAPM *Miscanthus* cellulose	25.0 ± 0.2
NAPM oat hull cellulose	25.8 ± 0.2
CPM *Miscanthus* cellulose	27.5 ± 0.2
CPM oat hull cellulose	28.1 ± 0.2

HTBT—hydrothermobaric treatment; DNAT—dilute nitric-acid treatment; AD—alkaline delignification; NAPM—nitric-acid pulping method; CPM—combined pulping method.

**Table 4 polymers-11-01645-t004:** Results of bacterial cellulose (BC) biosynthesis in enzymatic hydrolyzates produced from *Miscanthus* and oat-hull pretreatment products.

Sample	BC Biosynthesis
HTBT *Miscanthus*	–
HTBT oat hulls	–
DNAT *Miscanthus*	+
DNAT oat hulls	+
AD *Miscanthus*	+
AD oat hulls	+
NAPM *Miscanthus* cellulose	+
NAPM oat hull cellulose	+
CPM *Miscanthus* cellulose	+
CPM oat hull cellulose	+

HTBT—hydrothermobaric treatment; DNAT—dilute nitric-acid treatment; AD—alkaline delignification; NAPM—nitric-acid pulping method; CPM—combined pulping method; «–» BC hydrogel is absent; «+» BC hydrogel is present.

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
