# Peer review of "Pretreatments of Non-Woody Cellulosic Feedstocks for Bacterial Cellulose Synthesis"

_polymers, 2019, doi:10.3390/polym11101645_

Round 1
Reviewer 1 Report
This manuscript studies how the pretreatments of cellulosic natural materials affect the enzymatic hydrolysis and the formation of bacterial cellulose. These studies provide some valuable results on experimental operations, but lack of deep understanding on the fundamental mechanism. For example, the enzymatic hydrolysis is significantly improved after pre-treatments, is this caused by the chemical component changes or physical structure changes? What does the formation of BC depend on? Without these understandings, the present works just provide incremental results on the pretreatment optimization. Overall, I recommend major revisions for the current manuscript.
Could author add a flowchart to describe the pre-treatment routes? How about the product yields for various types of pre-treatments? Also, does the crystallinity of cellulose change during pre-treatments? Figure 1: Besides chemical components, I am wondering that whether the morphologies and porosity of the raw materials change after pre-treatments? How will these physical changes affect the biodegradation and synthesis of BC? Tables 2 and 3, keep one decimal place is good enough for glucose concentration. Line 404: The fact that no BC was formed in nutrient broths prepared from HTBT pretreatment products can be explained by inhibitors present in microquantities. Which types of inhibitors? There is no foreign element during HTBT treatments. For BC, how about the yields of BCs for miscanthus and oat hulls pretreated under various conditions? Figure 3: scale bars are not clear.
Reviewer 2 Report
The authors presented an interesting topic on the efficiency of various enzymatic treatments on non-woody plants and the feasibility of utilizing these enzymatic hydrolyzates for biosynthesis of bacterial cellulose. The various treatments and the results were well described and analyzed. The manuscript was well written.
The following are the detailed comments:
Line 47-53, It seems the authors only put the references in some, but not in all listed methods or pretreatment chemicals used. Line 212, please rephrase it for easy understanding Line 272, is it “%” or “times”? please verify Line 288, spelling mistake “rpoducts” Line 386, what is glucose concentrations for acetate-buffer hydrolyzates? It is advised to show in the sentence or the Table 3.
Round 2
Reviewer 1 Report
Figure 4: Scale bars are unclear, please modify them.
Does the Lignin content prohibit the formation of BC hydrogel?
